# C-Reactive Protein to Albumin Ratio as Prognostic Marker in Locally Advanced Non-Small Cell Lung Cancer Treated with Chemoradiotherapy

**DOI:** 10.3390/biomedicines10030598

**Published:** 2022-03-03

**Authors:** Alina Frey, Daniel Martin, Louisa D’Cruz, Emmanouil Fokas, Claus Rödel, Maximilian Fleischmann

**Affiliations:** 1Department of Radiation Oncology, Hospital of the Johann Wolfgang Goethe University, 60590 Frankfurt, Germany; alina.frey@kgu.de (A.F.); daniel.martin@kgu.de (D.M.); dcruzla@gmail.com (L.D.); emmanouil.fokas@kgu.de (E.F.); clausmichael.roedel@kgu.de (C.R.); 2German Cancer Research Center (DKFZ), 69120 Heidelberg, Germany; 3German Cancer Consortium (DKTK), Partner Site Frankfurt am Main, 60590 Frankfurt, Germany; 4Frankfurt Cancer Institute, 60590 Frankfurt, Germany

**Keywords:** NSCLC, chemoradiotherapy, CRT, CAR, C-reactive protein, albumin, inflammation, prognostic, predictive, biomarker

## Abstract

Despite the implementation of consolidative immune checkpoint inhibition after definitive chemoradiotherapy (CRT), the prognosis for locally advanced non-small-cell lung cancer (NSCLC) remains poor. We assessed the impact of the C-reactive protein (CRP) to albumin ratio (CAR) as an inflammation-based prognostic score in patients with locally advanced NSCLC treated with CRT. We retrospectively identified and analyzed 52 patients with primary unresectable NSCLC (UICC Stage III) treated with definitive/neoadjuvant CRT between 2014 and 2019. CAR was calculated by dividing baseline CRP by baseline albumin levels and correlated with clinicopathologic parameters to evaluate prognostic impact. After dichotomizing patients by the median, univariate and multivariate Cox regression analyses were performed. An increased CAR was associated with advanced T-stage (*p* = 0.018) and poor performance status (*p* = 0.004). Patients with pre-therapeutic elevated CAR had significantly lower hemoglobin and higher leukocyte levels (hemoglobin *p* = 0.001, leukocytes *p* = 0.018). High baseline CAR was shown to be associated with worse local control (LPFS, *p* = 0.006), shorter progression-free survival (PFS, *p* = 0.038) and overall survival (OS, *p* = 0.022), but not distant metastasis-free survival (DMFS). Multivariate analysis confirmed an impaired outcome in patients with high CAR (LPFS: HR 3.562, 95% CI 1.294–9.802, *p* = 0.011). CAR is an easily available and independent prognostic marker after CRT in locally advanced NSCLC. CAR may be a useful biomarker for patient stratification to individualize treatment concepts.

## 1. Introduction

Non-small-cell lung cancer (NSCLC) impacts more than 2.2 million people annually, representing the most common cause of cancer-related death worldwide [1]. At the time of diagnosis, approximately one-third of patients are diagnosed with Union for International Cancer Control (UICC) stage III disease [2]. Concurrent platinum-based doublet chemoradiotherapy (CRT), followed by immune checkpoint inhibition (ICI) for patients without disease progression after CRT, remains the standard of care for patients with primary unresectable locally advanced NSCLC. Updated survival analyses from the placebo-controlled phase III PACIFIC trial presented at the 2021 ASCO Annual Meeting have shown that sequential ICI with durvalumab resulted in a durable benefit for progression-free (PFS) and overall survival (OS) at 5 years [3]. However, in this cohort, OS rates were 42.9% and 33.4% (stratified HR 0.72, 95% CI 0.59–0.89; median 47.5 vs. 29.1 months), and PFS rates were 33.1% and 19.0% (stratified HR 0.55, 95% CI 0.45–0.68; median 16.9 vs. 5.6 months), respectively [3], indicating that further efforts are necessary to improve outcomes and survival. In this context, accurate risk stratification based on robust biomarkers could be a strategy to further improve and individualize treatment concepts.

The interplay between host immune response and cancer progression has been studied intensively for decades [4,5,6]. Cancer triggers systemic inflammation with a corresponding increase in pro-inflammatory cytokines such as interleukin 6 (IL6) and tumor necrosis factor α (TNFα) [4,5,6]. Upon cytokine stimulation, acute phase proteins including the C-reactive protein (CRP) are synthesized in hepatocytes [7]. CRP activates the complement system, initiates humoral and cellular effector mechanisms of the innate immune system, and is a common and routinely measured marker of inflammation [8].

Inflammatory cytokines lead to increased capillary permeability [9] and subsequently to a loss of serum albumin into the interstitium [10], while altered protein synthesis capacity contributes to a decrease in albumin synthesis during inflammation [11]. Therefore, CRP, and thus immune activation, correlates reciprocally with serum albumin levels [12,13], which is generally considered a marker of malnutrition and liver function [10].

An increased CRP to albumin ratio (CAR) has already been reported to be significantly associated with worse survival in various malignancies, including head and neck cancers and colorectal, anal and bladder cancer [14,15,16,17]. Previous studies on CAR in NSCLC have focused on early-stage or primary resectable disease, palliative regimens and second-line treatment [18,19,20]. More recently, Yang et al. further strengthened the negative prognostic impact of elevated baseline CAR in a prospective analysis of a heterogeneous cohort of 387 NSCLC patients [21].

Similar to CAR, the Glasgow Prognostic Score (GPS) reflects on CRP and albumin levels and has been explored in various settings. In brief, the results were consistent with those of CAR, and elevated GPS was associated with poorer survival in patients with NSCLC [22,23,24].

We here investigated the correlation of pretherapeutic CAR as a biomarker for inflammation and malnutrition with clinicopathologic features and its prognostic value in patients with locally advanced and primary unresectable UICC stage III NSCLC treated with CRT. In addition, to account for GPS and to compare its prognostic value with that of CAR, we performed corresponding survival analyses.

## 2. Materials and Methods

### 2.1. Patients and Treatment Protocol

We retrospectively identified 52 patients with locally advanced and primary unresectable NSCLC (UICC III) treated between 1/2014 and 12/2019 with definitive or neoadjuvant-intended CRT. Pseudonymized data were used after institutional ethics committee approval.

Staging was routinely performed, including clinical examination and pulmonary function testing, computed tomography (CT) of the chest and abdomen and positron emission tomography/computed tomography (PET/CT). For brain metastases, screening magnetic resonance imaging (MRI) was performed. Histological confirmation and pathological classification were provided after bronchoscopic, thoracoscopic or CT-guided biopsy. The decision on definitive versus neoadjuvant-intended CRT and potential operability was made by a multidisciplinary tumor board, with consideration of the patient’s preferences.

Radiotherapy (RT) was applied using either 3D-conformal radiotherapy or intensity-modulated RT (IMRT). Patients were treated with a median total dose of 66.6 Gray (Gy) (range: 45–66.6 Gy) with daily fractions of 1.8 or 2 Gy five days per week. Patients primarily assigned to neoadjuvant-intended CRT were treated with a median dose of 50.4 Gy (range: 45–59.4 Gy). Except for one patient, all patients received RT as planned without dose reduction.

Concurrent chemotherapy consisted of platinum-based chemotherapy and vinorelbine. Platinum-based chemotherapy was administered in the first and fifth week of RT. Generally, cisplatin (20 mg/m^2^, day 1–5 and day 28–33) was preferred. For patients not eligible for cisplatin-based chemotherapy for various reasons (e.g., renal insufficiency), carboplatin (AUC1, day 1–5 and day 28–33) was applied. Vinorelbine (50 mg/m^2^, per os) was given on days 1, 8, 15, 29, 36 and 43. Etoposide (90 mg/m^2^, day 1–3 and day 28–31) was used instead of vinorelbine, e.g., in patients with neuroendocrine differentiation. Seventeen patients received induction chemotherapy before CRT. Induction chemotherapy had no effect on OS (*p* = 0.837); thus, these patients were included. In addition, no differences in OS were found between patients who received neoadjuvant-intended or definitive CRT (*p* = 0.758), so these patients were also included. Eight of nine patients who received neoadjuvant-intended treatment underwent surgery.

### 2.2. Response Assessment and Follow-Up

Initial response assessment and restaging procedures including pulmonary function testing and imaging (CT scans of chest and abdomen) were performed 6–8 weeks after completion of CRT. In the first year after the completion of therapy, follow-up examinations were scheduled every three months, afterwards every six or twelve months or inter-individually depending on the respective risk profile and clinical history.

### 2.3. Clinicopathologic Features and Serum Chemistry

Personal data and clinicopathologic features were retrieved from the patient files of our hospital database including sex and age, general physical condition at the time of diagnosis according to the Eastern Cooperative Oncology Group (ECOG) Performance Status, TNM stage, UICC stage and histology. Height and weight were documented at the beginning of the therapy; body mass index (BMI) was calculated by dividing weight in kilograms by height in meters squared (kg/m^2^). Blood parameters were obtained as part of routine diagnostics at the start of therapy. The baseline was either defined on the day of therapy initiation or up to 9 days before treatment.

CAR was calculated by dividing CRP in mg/dL through albumin in g/dL. GPS was determined according to Forrest et al. [25]. Patients with CRP values ≤ 1.0 mg/dL and albumin values ≥ 3.5 g/dL were categorized as GPS 0, CRP levels > 1.0 mg/dL or albumin levels < 3.5 g/dL as GPS 1, and CRP levels > 1.0 mg/dL and albumin levels < 3.5 g/dL as GPS 2. Leukocyte and platelet counts per nanoliter (nL), as well as hemoglobin in g/dL, were recorded regularly to define the patient-specific nadir (defined as lowest blood count from start to 4 weeks after completion of CRT).

### 2.4. Statistical Analysis

Differences between groups regarding continuous variables were tested for statistical significance via the Mann–Whitney U test for independent samples. Survival times were calculated from the start of CRT to the date of respective events or last follow-up. Events were defined as death from any cause (overall survival, OS), and progression was defined as local recurrence/progression after remission/stable disease, or distant metastasis (progression-free survival, PFS), isolated local recurrence/progression (local progression-free survival, LPFS), or distant metastasis (distant metastasis-free survival, DMFS). Survival time analyses were plotted according to the Kaplan–Meier method and the log-rank test was used for calculation.

We further performed multivariate Cox analyses with the calculation of hazard ratios (HRs) and corresponding 95% confidence intervals (CIs) to investigate the influence of high or low CAR (cut-off: median) and other categorical predictor variables (cut-off for not already dichotomous variables: median, standard values or laboratory limits) on survival. For reliable assessment of the results of multivariate Cox analysis, at least ten events should occur for each prognostic variable included [26]. Due to the low number of events, we only included prognostic variables in the multivariate model that were statistically significant in the univariate Cox analysis. All statistical analysis was performed using IBM SPSS Version 27 (Armonk, NY, USA). All tests were two-sided, and a *p*-value of 0.05 was considered statistically significant.

## 3. Results

### 3.1. Patient Characteristics

We retrospectively assessed 52 patients (female *n* = 27, 51.9%) with locally advanced and primary unresectable UICC stage III NSCLC (T1—4, N0—3, M0). Median age was 66 (range: 47–79) years. Median follow-up was 17 (range: 2–76) months. Twenty-eight of fifty-two patients (53.8%) had a CAR ≤ the median. Detailed pre-treatment patient characteristics are summarized in Table 1.

Higher T-stage (T1/2 vs. T3/4, *p* = 0.018) and poorer performance status (≥ECOG 1, *p* = 0.004) were significantly associated with elevated baseline/pre-treatment CAR, whereas N-status (N0/1 vs. N2/3), sex, BMI (cut-off: 25 kg/m^2^) and histology (adeno and squamous cell) were not (*p* = 0.265, *p* = 0.763, *p* = 0.971 and *p* = 0.155) (Figure 1A).

We examined associations between CAR as a continuous variable with baseline blood count and blood count at the patient-specific nadir. Significantly higher leukocyte counts (cut-off: 10/nL, *p* = 0.018) and lower hemoglobin levels (sex-specific cut-off for men: 13 g/dL, cut-off for women: 12 g/dL, *p* = 0.001) were found at baseline in patients with a higher CAR. Platelet count at baseline showed no significant association with CAR (cut-off: 400/nL, *p* = 0.241). In the patient-specific nadir, a more profound treatment-related leukocyte drop (cut-off: median, 6.2/nL, *p* = 0.001), lower hemoglobin (cut-off: median, 10 g/dL, *p* = 0.001) and thrombocyte values (cut-off: median, 144/nL, *p* = 0.017) were associated with a higher CAR (Figure 1B–D).

A higher baseline CAR was significantly associated with an advanced T-stage, poorer performance status (A), lower baseline hemoglobin values and higher baseline leukocyte counts. In the patient-specific nadir, elevated CAR was correlated to lower hemoglobin and thrombocyte levels and a stronger decrease in the leukocyte count.

### 3.2. Clinical Outcomes: Disease Control and Survival

#### 3.2.1. Univariate Analysis

The OS and PFS at 3 years were 33% and 18%, respectively. We have dichotomized the cohort at the median CAR of 0.32 to perform the survival analyses. Elevated baseline CAR (> median) was significantly associated with worse OS (*p* = 0.022), PFS (*p* = 0.038) and LPFS (*p* = 0.006), but not with DMFS (*p* = 0.207) (Figure 2). The results of the Cox analysis are given in Table 2. In the univariate Cox analysis, only lower leukocyte levels in the patient-specific nadir (cut off: median, 2.3/nL, HR 0.257, 95% CI 0.084–0.793, *p* = 0.011) and male sex (HR 2.711, 95% CI 0.979–7.508, *p* = 0.046) were significantly associated with worse LPFS (Appendix A).

Elevated baseline CAR was significantly associated with worse OS (A), LPFS (B) and PFS (C), but not DMFS (D). The number of patients at risk is given below.

In a next step, we created a subgroup excluding non-adenocarcinomas and non-squamous cell carcinomas to further homogenize the patient cohort and to consider potential confounding factors. In this cohort, a high pre-therapeutic CAR was confirmed as significantly associated with an impaired OS (HR 2.268, 95% CI 1.086–4.736, *p* = 0.025), PFS (HR 2.105, 95% CI 1.031–4.295, *p* = 0.037) and LPFS (HR 4.426, 95% CI 1.493–13.121, *p* = 0.004). Furthermore, in this subgroup, patients with squamous cell carcinoma (HR 2.456, 95% CI 1.178–5.122, *p* = 0.013) and a high BMI (cut-off: 25 kg/m^2^, HR 2.161, 95% CI 1.014–4.604, *p* = 0.041) had a shorter OS. Poorer locoregional control was found in men (HR 3.303, 95% CI 1.097–9.943, *p* = 0.025), squamous cell carcinomas (HR 3.826, 95% CI 1.304–11.225, *p* = 0.009) and low nadir leukocyte levels (HR 0.200, 95% CI 0.056–0.712, *p* = 0.006). Univariate Cox regression analyses with corresponding outcome variables are given in Appendix A. Interestingly, GPS was not significantly associated with OS, PFS, LPFS or DMFS (*p* = 0.441, *p* = 0.634, *p* = 0.222, *p* = 0.800) (Appendix A).

#### 3.2.2. Multivariate Analysis

Multivariate Cox regression analyses were only performed for LPFS and OS, as only outcome parameters that could be predicted by other variables in addition to CAR were listed. For PFS, CAR was the only significant predictor variable and DMFS could not be predicted by CAR in either subgroup. Stepwise backward elimination was chosen as the model for the multivariate analyses. Consequently, all independent variables that significantly predicted the outcome in univariate Cox regression analysis (Appendix A) were first included in the model and then successively removed if they did not contribute significantly. Values above *p* > 0.1 were set as exclusion criteria for the variables in the model.

In the multivariate analyses, only CAR remained predictive for worse OS, whereas CAR and nadir leukocyte count remained predictive for adverse LPFS (CAR, cut-off: 0.32, median, HR 3.562, 95% CI 1.294–9.802, *p* = 0.011; nadir leukocyte count, cut-off: 2.3/nl, median, HR 0.266, 95% CI 0.085–0.836, *p* = 0.013) (Table 3).

Excluding non-adenocarcinomas and non-squamous cell carcinomas revealed a significant correlation of an elevated baseline CAR with a worse OS (HR 2.480, 95% CI 1.178–5.220, *p* = 0.018). BMI (cut-off: 25 kg/m^2^, HR 2.359, 95% CI 1.103–5.045, *p* = 0.024) remained statistically significant (Appendix A).

## 4. Discussion

All stages of tumorigenesis are closely intertwined with inflammation [4]. Chronic inflammation contributes to tumor development by influencing the extracellular matrix, the tumor microenvironment and neoangiogenesis, thus promoting tumor growth and metastatic behavior [27]. CRP secreted by the liver upon IL6 stimulation is a dynamic laboratory parameter easily accessible for detecting and monitoring acute and chronic systemic inflammation as well as tumorigenic inflammatory response. It has been reported to be associated with impaired outcomes in oral and oropharyngeal cancer; gastrointestinal malignancies, including colorectal, pancreatic and hepatocellular cancer; as well as urological malignancies [28,29,30,31,32]. The prognostic impact of elevated CRP levels in patients with NSCLC has been investigated in various settings and several studies. In their meta-analysis of 1649 patients, Jin et al. analyzed eight studies and showed an association between elevated CRP levels and worse survival in patients with both primary resectable NSCLC and primary unresectable NSCLC [33].

Several more specific inflammatory scores based on cellular components of the blood have been explored for predicting response and survival in NSCLC [34,35]. For example, a high post-CRT neutrophil-to-lymphocyte ratio (NLR) was shown to be associated with significantly worse LPFS and OS in patients with locally-advanced NSCLC treated with definitive CRT [36,37]. Neutrophils can be stimulated by the tumor itself and—depending on their phenotypic and functional polarization—are considered tumor-promoting [38]. The neutrophil-to-lymphocyte ratio may represent the balance between pro-tumoral inflammatory status and anti-tumoral immune response. In addition to NLR, a lower lymphocyte-to-monocyte ratio (LMR) has also been associated with an impaired prognosis in NSCLC patients [39]. Interestingly, an increasing LMR after nivolumab was related to an improved PFS in a cohort of 75 NSCLC patients, indicating LMR as a dynamic surrogate marker for response [40]. However, lymphocytopenia is a common side effect of radiation [41]. Therefore, a prognostic score that depends on the lymphocyte count could be considerably biased in patients who have undergone radiotherapy.

In contrast, serum albumin levels are a parameter of malnutrition and decrease during a systemic state of inflammation through increased CRP synthesis. The prevalence of malnutrition, cachexia and wasting, characterized by a significant weight loss, sarcopenia and decline in skeletal muscles in cancer patients, ranges from 30 to 80% and increases in terminal stages [42]. These facts indicate a strong interplay between nutrition, sarcopenia, inflammation and cancer progression. Thus, low albumin levels and nutrient indices have been shown to be associated with poorer outcomes in survival time analyses in various malignancies [43,44,45]. In two smaller cohorts, hypoalbuminemia has been shown to be a negative and independent prognostic factor in patients with metastatic or non-metastatic NSCLC initially treated with chemotherapy or targeted therapies [45,46]. In addition, Rim et al. studied 353 elderly cancer patients treated with definitive RT and showed that low albumin levels were associated with a higher rate of poor treatment compliance [47]. In general, the state of systemic inflammation and worse nutritional status is often associated with higher tumor stages [48] and inferior survival in cancer patients [28,29,30,31,32,33,43,44,45], suggesting the potential of combined scores.

In the present study, we have assessed CAR as a prognostic and easily accessible biomarker in a homogenous cohort of 52 UICC stage III NSCLC patients treated uniformly with CRT. First, we have demonstrated that higher baseline CAR was significantly associated with the higher T stage (T1/2 vs. T3/4) and poor performance status (ECOG 0 vs. ECOG 1/2). Second, baseline CAR was significantly correlated to pre-treatment blood counts and the patient-specific nadir during CRT. Ultimately, elevated baseline CAR was associated with an impaired LPFS, PFS and OS. In multivariate analysis, CAR was shown to be an independent predictor of LPFS in the entire cohort. After exclusion of all non-squamous cell carcinomas and non-adenocarcinomas, CAR remained significant for OS. These results are consistent with previous studies and strengthen the prognostic impact of CAR [18,19,20,21].

As GPS was also reported as a prognostic marker for survival, we performed corresponding analyses for this score. However, GPS did not predict survival in our cohort. In contrast, Yotsukura et al. explored the prognostic impact of GPS in a large cohort of 1048 patients with primary resectable NSCLC. A high GPS was significantly associated with poor OS [22]. Kishi et al. reported GPS as a significant prognostic factor in a cohort of 165 patients with early-stage NSCLC treated with stereotactic body radiation therapy (SBRT) [23]. In a heterogeneously treated cohort of 261 unresectable NSCLC patients, high GPS was associated with poorer ECOG performance status and increased baseline leukocyte count. Moreover, GPS was associated with worse cancer-specific survival (CSS) [24]. A meta-analysis by Zhu et al., including 2669 patients from 12 studies, confirmed GPS as a predictor of survival in patients with NSCLC [49]. Interestingly, another meta-analysis by Jin et al., which evaluated the prognostic impact of GPS in 5817 patients from 11 studies, failed to show an association between GPS and worse survival among patients who underwent surgery [50]. In our cohort, we found albumin levels < 3.5 g/dL in three patients only. Consistent with the findings of Matsumoto et al. [51], CAR appears to be a more reliable prognostic marker than GPS in cohorts where few patients have notably low albumin levels.

Unlike previous studies in which a higher BMI was associated with better survival rates [52], high BMI was associated with an inferior outcome in this study. One possible explanation is that patients with a high BMI in our cohort, e.g., in the context of metabolic syndrome, have known or initially undiagnosed preexisting conditions that have an unfavorable impact on the outcome. In addition, BMI does not necessarily reflect body composition and sarcopenia, which may have a greater impact on outcome and survival. However, in our cohort, only two patients were underweight (BMI < 18.5 kg/m^2^) and five were obese (BMI ≥ 30 kg/m^2^), so caution should be used when interpreting these results. Adenocarcinomas were associated with prolonged OS, consistent with previous studies [53]. Furthermore, women had a better outcome, which is in line with other studies [1,54]. Low leukocyte levels during therapy, which may indicate increased toxicity [55], were associated with shorter survival in our cohort. As a result, a reduction in the chemotherapy dose may have been indicated, affecting disease control and survival.

Nevertheless, there are several limitations in the present study. Due to the retrospective design and the small sample size, confounding and sampling bias cannot be excluded. The results of multivariate Cox regression analysis require caution because of the relatively small number of events that occurred. We chose the median as the cut-off of the CAR. Previous studies using other statistical methods (e.g., receiver operating characteristic (ROC) curves) to determine cut-off values for CAR arrived at cut-off values from 0.0271 to 0.83 [14,15,16,17,18,19,20,21]. Further prospective studies could clarify which cut-off is most appropriate for CAR.

In summary, CAR is an easily accessible biomarker that reflects systemic inflammation and nutritional status and was associated with advanced tumor stage (T-stage), poor performance status and poor survival in patients with UICC stage III NSCLC. These findings may indicate that CAR is a tool for risk stratification and highlight the importance of early integration of interdisciplinary supportive care to improve treatment compliance and prognosis.

## Figures and Tables

**Figure 1 biomedicines-10-00598-f001:**
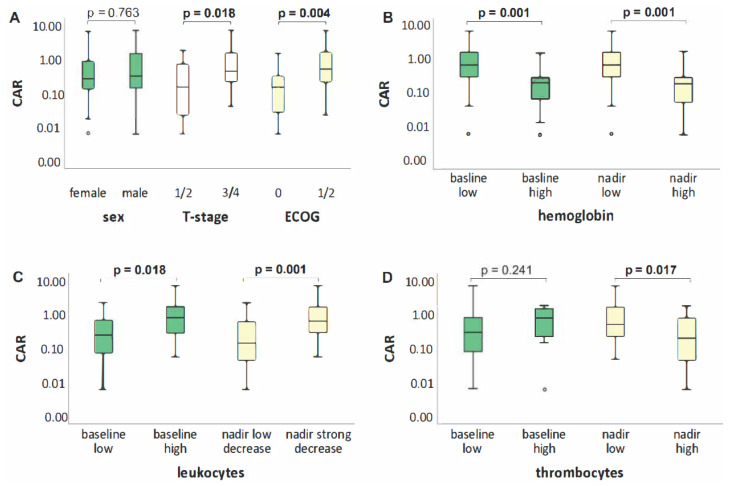
Differences in CAR by sex, T-stage, ECOG (**A**) and blood count (hemoglobin (**B**). leukocytes (**C**) and thrombocytes (**D**).

**Figure 2 biomedicines-10-00598-f002:**
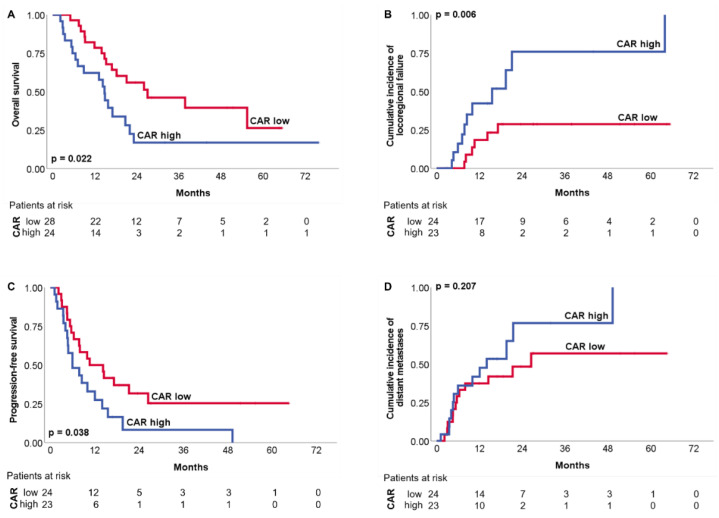
OS (**A**), LPFS (**B**), PFS (**C**) and DMFS (**D**) according to dichotomized CAR.

**Table 1 biomedicines-10-00598-t001:** Patient, disease, treatment, blood characteristics.

			Median or *n* (Range or %)
Patients	Sex	Male	25 (48.1)
		Female	27 (51.9)
	Age (years)		66 (47–79)
	BMI (kg/m^2^)		25.14 (16.48–39.12)
	ECOG	0	16 (30.8)
		1	30 (57.7)
		2	6 (11.5)
Tumor	T-stage	T1	4 (7.7)
		T2	14 (26.9)
		T3	16 (30.8)
		T4	18 (34.6)
	N-stage	N0	6 (11.5)
		N1	2 (3.9)
		N2	26 (50.0)
		N3	18 (34.6)
	Histology	Adeno	29 (55.8)
		Squamous cell	17 (32.7)
		Spindle cell (sarcomatoid)	1 (1.9)
		Neuroendocrine (large cell)	3 (5.8)
		NOS	2 (3.8)
Treatment	CRT	Definitive	43 (82.7)
		Neoadjuvant-intended	9 (17.3)
	Induction CT	Yes	17 (32.7)
		No	35 (67.3)
	Total dose (Gy)		66.6 (45–66.6)
Baseline	CRP (mg/dL)		1.37 (0.03–21.43)
	Albumin (g/dL)		4.1 (2.8–4.9)
	CAR		0.32 (0.01–7.14)
	Hemoglobin (g/dL)		12.35 (8.0–15.2)
	Leukocytes (/nL)		8.39 (3.29–80.95)
	Thrombocytes (/nL)		312.5 (117–679)
	GPS	0	20 (38.5)
		1	29 (55.8)
		2	3 (5.8)
Nadir	Hemoglobin (g/dL)		10.0 (6.0–12.3)
	Leukocytes (/nL)		2.28 (0.15–7.13)
	Thrombocytes (/nL)		144 (10–305)

Abbreviations: BMI, body mass index; ECOG, Eastern Cooperative Oncology Group Performance Status; NOS, not otherwise specified; CRT, chemoradiotherapy; CT, chemotherapy; Gy, Gray; CRP, C-reactive protein; CAR, CRP to albumin ratio; GPS, Glasgow Prognostic Score.

**Table 2 biomedicines-10-00598-t002:** Univariate Cox regression analysis with dichotomized CAR.

	HR	95% CI	*p*-Value
OS	2.178	1.101–4.310	**0.022**
PFS	2.005	1.026–3.918	**0.038**
LPFS	3.723	1.365–10.151	**0.006**
DMFS	1.658	0.749–3.671	0.208

Abbreviations: OS, overall survival; PFS, progression-free survival; LPFS, local progression-free survival; DMFS, distant metastasis-free survival.

**Table 3 biomedicines-10-00598-t003:** Multivariate Cox regression analysis.

		HR	95% CI	*p*-Value
LPFS	CAR	3.562	1.294–9.802	**0.011**
	Leukocytes nadir	0.266	0.085–0.836	**0.013**
	Sex	removed		

OS was predicted by CAR only. Abbreviations: LPFS, local progression-free survival; CAR, C-reactive protein to albumin ratio.

## Data Availability

Data are available upon reasonable request.

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
