# Peer review of "C-Reactive Protein to Albumin Ratio as Prognostic Marker in Locally Advanced Non-Small Cell Lung Cancer Treated with Chemoradiotherapy"

_biomedicines, 2022, doi:10.3390/biomedicines10030598_

Round 1

Reviewer 1 Report

Traditionally, the neutrophil to lymphocyte ratio (NLR) or the monocyte to lymphocyte ratio (MLR) is used to assess systemic inflammation. In the article submitted for review, the ratio of CRP to albumin is proposed. From a methodological point of view, I have no questions about the study. However, I would like to clarify with the authors why they did not calculate the traditionally used indices? Maybe NLR or MLR would have had better prognostic characteristics, or their combination with CRP to albumin ratio would have provided more valuable prognostic information. I think it is appropriate to exclude patients with non-NSCLC histology from the study group. 

Author Response

Dear Editors,

Dear Reviewers,

Thank you for the opportunity to resubmit a revised version of our manuscript. In addition, we would like to thank you for the time and effort you have put into your valuable and helpful review.

We have implemented your constructive feedback and suggestions to the best of our ability to improve our manuscript.

Below you will find the point-by-point response to the reviewers' comments. Changes in the manuscript and figures are marked in yellow.

Once again, we highly appreciate your profound and objective review and thank you for your time and effort.

Maximilian Fleischmann

Comments for the Author: Traditionally, the neutrophil to lymphocyte ratio (NLR) or the monocyte to lymphocyte ratio (MLR) is used to assess systemic inflammation. In the article submitted for review, the ratio of CRP to albumin is proposed. From a methodological point of view, I have no questions about the study. However, I would like to clarify with the authors why they did not calculate the traditionally used indices? Maybe NLR or MLR would have had better prognostic characteristics, or their combination with CRP to albumin ratio would have provided more valuable prognostic information. I think it is appropriate to exclude patients with non-NSCLC histology from the study group. 

Response to the Reviewer: Thank you for your kind review. You have raised an interesting point. Indeed, neutrophil to lymphocyte ratio (NLR) is a widely investigated prognostic score in various malignancies and has been assessed for NSCLC in different settings (PMIDs: 33634385; 33569306; 33999381; 34632714). Neutrophil to lymphocyte ratio may represent the balance between pro-tumoral inflammatory status and anti-tumoral immune response. In contrast, the CRP to albumin ratio has been less investigated and was therefore evaluated in this study. CRP to Albumin ratio appears to be a rather unspecific score that also reflects the nutritional status of patients and correlates to some extent with overall performance status. In addition, lymphocytopenia is a common side effect of radiation and can be observed in many patients. Therefore, especially when using lymphocytes, the results may be biased, which reduces the validity compared to other prognostic scores. Please indulge us if we do not discuss other prognostic scores in this publication. We are trying to keep the manuscript focused and concise.

We agree with you to exclude non-squamous cell carcinomas and non-adenocarcinomas from the study. Because of the relatively small number of patients, we therefore performed corresponding subgroup analyses [OS (HR 2.268, 95% CI 1.086 – 4.736, p = 0.025), PFS (HR 2.105, 95% CI 1.031 – 4.295, p = 0.037) and LPFS (HR 4.426, 95% CI 1.493 – 13.121, p = 0.004)].

Once again, thank you very much for your review. We hope you understand that we want to keep our manuscript concise and do not specifically address the NLR and MLR.

Yours sincerely

MF

Reviewer 2 Report

This manuscript, written by Dr. Alina Frey et al., original research, with the title of “C-Reactive Protein to Albumin Ratio as Prognostic Marker in Locally Advanced Non-Small Cell Lung Cancer Treated with Chemoradiotherapy” analyzed the prognostic value of CAR is a series of 52 patients with locally advanced Non-small cell lung cancer (NSCLC). They found that CAR was a relevant predictor factor.

This manuscript is well written, easy to read and to understand. Since there are many abbreviations, I would recommend listing them at the beginning or at the end. I would also recommend showing the raw results of all univariate analyses for the different prognostic variables (OS, PFS, etc) in a supplementary table. Additionally, there are a series of comments that may help improve the manuscript:

(1) Introduction. Regarding the sentences “Concurrent platinum-based doublet chemoradiotherapy (CRT), followed by immune checkpoint inhibition (ICI) for patients without disease-progression after CRT and programmed death-ligand 1 (PD-L1) combined positivity score (CPS) ≥ 1%, remains the standard of care for patients with primary unresectable locally advanced NSCLC.”

Could you please confirm that the use of CPS is correct in this context?

Could you please confirm about the tumor proportion score (TPS)?

(2) Introduction. Regarding the III PACIFIC trial. Could you please add this words in the text to highlight that it is communication and not a manuscript? For example, “Meeting Abstract (2021 ASCO Annual Meeting).”

(3) Introduction. Regarding the III PACIFIC trial information. Could you please add the p values for the OS and PFS curves?

(4) Introduction. Regarding CRP. Could you please add that “CRP Polarizes Human Macrophages to a M1 Phenotype and Inhibits Transformation to the M2 Phenotype”? I think this is relevant as M1 polarization, which in theory forms part of the host immune response to cancer (PMID: 21415385).

(5) Introduction. Regarding albumin. Could you please add “It was previously noted that acute-phase concentrations of proteins, such as CRP, tend to rise in inflammatory conditions, while albumin concentrations tend to decline” (PMID: 34181998).

(6) Introduction. Regarding the GPS. Could you please describe how it is calculated?

For example, patients who had both a serum elevation of CRP (>1.0 mg/dL) and hypoalbuminemia (<3.5 g/dL) were allocated a GPS of 2. Patients with only one of the abnormal values were allocated a GPS of 1, and patients who had neither were allocated a GPS of 0 (PMID: 25216413).

(7) Introduction, last paragraph. Regarding “CRT.” Do it mean consolidative immune checkpoint inhibition after definitive chemoradiotherapy (CRT) “?

(8) Page 3 of 11. First paragraph. Which was the criteria used for using neoadjuvant-intended or definitive CRT treatments? Which is the difference between both types of treatment?

(9) Page 3 of 11. Could you please describe the meaning of “patient-specific nadir”?

(When discussing chemotherapy adverse reactions often you will hear the word nadir, mainly in reference to the blood counts, particularly white blood cell count and platelet count. Nadir means low point, however further explanation may clarify this term in connection with chemotherapy treatment).

(10) Page 3 of 11. It is stated that the median was used to differentiate high vs low CAR. Is this the standard cut-off strategy for this marker? Do other groups use the same cut-off?

(11) Page 3 of 11. Regarding SPSS. Could you please add “Armonk, New York, USA”?

(12) Table 1. Regarding the histological classification.

Could you please use the nomenclature as shown in the current 5th revised WHO classification? I.e. adenocarcinoma, squamous cell carcinoma, spindle cell carcinoma (pleomorphic carcinoma, sarcomatoid carcinoma), neuroendocrine carcinomas (small or large cell?).

NOS? Not-otherwise–specified?

(13) Regarding section 3.1. and Table 1. Patient’s characteristics. Could you please specify/confirm the exact time that these variables were assessed? Are all the values at diagnosis and before any treatment?

(14) Page 5 of 11. Can you please describe the meaning of “baseline CAR”? (It may help the reader who is not familiar/specialist in this field).

(15) Regarding Figure 1. Why CAR is evaluated as a continuous variable in this case, and not as a dichotomic using the median cut-off?

(16) Since the OS and PFS analysis uses the CAR high/low, could you please correlate CAR high/low with the other clinicopathological variables as shown in table 1? Including the p values? Or do you think that the information present as a continuous CAR variable is enough?

(17) Regarding the multivariate analysis, I understand that you performed a COX for OS, with the method of backward conditional?

(18) In Table 3. “Removed” means removed in the process of backward conditional?

Independently of the other variables. Did anti-PD-L1 therapy improved the survival of the patients?

(19) Regarding the CAR variable. Which component of it (CRP or albumin) was the most relevant for the prognosis of the patients?

(20) If CRP is associated to antitumor host immune response, how do you interpret that high CAR associated to poor prognosis? Or CRP is reflecting other immune system status?

Author Response

Dear Editors,

Dear Reviewers,

Thank you for the opportunity to resubmit a revised version of our manuscript. In addition, we would like to thank you for the time and effort you have put into your valuable and helpful review.

We have implemented your constructive feedback and suggestions to the best of our ability to improve our manuscript.

Below you will find the point-by-point response to the reviewers' comments. Changes in the manuscript and figures are marked in yellow.

Once again, we highly appreciate your profound and objective review and thank you for your time and effort.

Maximilian Fleischmann

Comments for the Author: This manuscript, written by Dr. Alina Frey et al., original research, with the title of “C-Reactive Protein to Albumin Ratio as Prognostic Marker in Locally Advanced Non-Small Cell Lung Cancer Treated with Chemoradiotherapy” analyzed the prognostic value of CAR is a series of 52 patients with locally advanced Non-small cell lung cancer (NSCLC). They found that CAR was a relevant predictor factor.

This manuscript is well written, easy to read and to understand. Since there are many abbreviations, I would recommend listing them at the beginning or at the end. I would also recommend showing the raw results of all univariate analyses for the different prognostic variables (OS, PFS, etc) in a supplementary table. Additionally, there are a series of comments that may help improve the manuscript:

Response to the Reviewer: We highly appreciate your extremely detailed, profound and kind review! We have added Supplementary Table S2 including the univariate analysis of all variables. Moreover, we have created an overview of all abbreviations at the beginning of the manuscript.  

(1) Introduction. Regarding the sentences “Concurrent platinum-based doublet chemoradiotherapy (CRT), followed by immune checkpoint inhibition (ICI) for patients without disease-progression after CRT and programmed death-ligand 1 (PD-L1) combined positivity score (CPS) ≥ 1%, remains the standard of care for patients with primary unresectable locally advanced NSCLC.”

Could you please confirm that the use of CPS is correct in this context?

Could you please confirm about the tumor proportion score (TPS)?

Response to the Reviewer: Certainly, you have addressed a controversial point: In Germany/Europe durvalumab is administered after chemoradiation to patients with a combined positivity score (CPS) ≥ 1%, based on a post hoc analysis that failed to show an OS benefit in patients with PD-L1 expression less than 1%. The U.S. is rather agnostic about PD-L1 expression. However, we have rephrased the passage so that it is now a bit more general.

(2) Introduction. Regarding the III PACIFIC trial. Could you please add this words in the text to highlight that it is communication and not a manuscript? For example, “Meeting Abstract (2021 ASCO Annual Meeting).”

Response to the Reviewer: Of course, thanks for pointing this out.

(3) Introduction. Regarding the III PACIFIC trial information. Could you please add the p values for the OS and PFS curves?

Response to the Reviewer: We have added the corresponding hazard ratios (HR) from the ASCO meeting abstract. For more information, investigators from PACIFIC trial have recently published an updated 4-year survival analysis (PMID: 33476803)

(4) Introduction. Regarding CRP. Could you please add that “CRP Polarizes Human Macrophages to a M1 Phenotype and Inhibits Transformation to the M2 Phenotype”? I think this is relevant as M1 polarization, which in theory forms part of the host immune response to cancer (PMID: 21415385).

Response to the Reviewer: We agree with you that this is a potentially interesting point. Nonetheless, we briefly addressed the activation of humoral and cellular immune response. Therefore, we do not intend to emphasize the complex role of single components of the immune response. M2 macrophages are discussed to have cancer promoting capabilities including immuno-suppression, angiogenesis and neovascularization, as well as stromal activation and remodeling, while M1 macrophages (promoted by CRP) are more likely to be associated with anti-tumor response. Moreover, there are several factors influencing macrophage polarization and this would considerably exceed the extent of an introduction.

(5) Introduction. Regarding albumin. Could you please add “It was previously noted that acute-phase concentrations of proteins, such as CRP, tend to rise in inflammatory conditions, while albumin concentrations tend to decline” (PMID: 34181998).

Response to the Reviewer: Correct. We have already described this fact and included the excellent manuscript you suggested in our references.

(6) Introduction. Regarding the GPS. Could you please describe how it is calculated?

For example, patients who had both a serum elevation of CRP (>1.0 mg/dL) and hypoalbuminemia (<3.5 g/dL) were allocated a GPS of 2. Patients with only one of the abnormal values were allocated a GPS of 1, and patients who had neither were allocated a GPS of 0 (PMID: 25216413).

Response to the Reviewer: You may have overlooked it. GPS was described in “Materials and Methods”. “GPS was determined according to Forrest et al. [24]. Patients with CRP values ≤ 1.0 mg/dl and albumin values ≥ 3.5 g/dl were categorized as GPS 0, CRP levels > 1.0 mg/dl or albu-min levels < 3.5 g/dl as GPS 1, and CRP levels > 1.0 mg/dl and albumin levels < 3.5 g/dl as GPS 2.”

(7) Introduction, last paragraph. Regarding “CRT.” Do it mean consolidative immune checkpoint inhibition after definitive chemoradiotherapy (CRT) “?

Response to the Reviewer: This passage may seem misleading. The study presented here refers mainly to the response and outcome after chemoradiation (CRT). Only a negligible proportion of patients have received consolidative therapy with immune checkpoint inhibitors (ICI), whereas consolidative ICI (no disease progression; CPS ≥ 1%) with durvalumab is now standard after CRT.

(8) Page 3 of 11. First paragraph. Which was the criteria used for using neoadjuvant-intended or definitive CRT treatments? Which is the difference between both types of treatment?

Response to the Reviewer: Good point! We have included the following passage in the manuscript which should make clear the differences in the respective treatment approach. “The decision on definitive versus neoadjuvant-intended CRT and potential operability was made by a multidisciplinary tumor board, with consideration of the patient's preferences.”

(9) Page 3 of 11. Could you please describe the meaning of “patient-specific nadir”?

(When discussing chemotherapy adverse reactions often you will hear the word nadir, mainly in reference to the blood counts, particularly white blood cell count and platelet count. Nadir means low point, however further explanation may clarify this term in connection with chemotherapy treatment).

Response to the Reviewer: Exactly! We have clarified the definition of nadir as lowest blood count from start to 4 weeks after the completion of CRT.

(10) Page 3 of 11. It is stated that the median was used to differentiate high vs low CAR. Is this the standard cut-off strategy for this marker? Do other groups use the same cut-off?

Response to the Reviewer: Very interesting point! The median is generally a robust statistical value that is less subject to deviations. Certainly, it represents the distribution of variables only imprecisely. However, to compare two categorical variables, the median should still be of acceptable efficiency. If CAR is wanted to be a continuous variable for the survival time analysis, the cut-off can be determined using maximally selected rank statistics. This method, however, showed no difference to the median in our analysis and therefore we limited the analysis to the median as a common value.

(11) Page 3 of 11. Regarding SPSS. Could you please add “Armonk, New York, USA”?

Response to the Reviewer: Added!

(12) Table 1. Regarding the histological classification.

Could you please use the nomenclature as shown in the current 5th revised WHO classification? I.e. adenocarcinoma, squamous cell carcinoma, spindle cell carcinoma (pleomorphic carcinoma, sarcomatoid carcinoma), neuroendocrine carcinomas (small or large cell?).

NOS? not otherwise specified?

Response to the Reviewer: Agreed. We checked the corresponding patient and added the information (sarcomatoid). All neuroendocrine differentiated carcinomas were large cell. NOS (not otherwise specified) is explained under the table in the list of abbreviations.

(13) Regarding section 3.1. and Table 1. Patient’s characteristics. Could you please specify/confirm the exact time that these variables were assessed? Are all the values at diagnosis and before any treatment?

Response to the Reviewer: Table 1 summarizes the baseline characteristics of the cohort. We have added the information (marked in yellow).

(14) Page 5 of 11. Can you please describe the meaning of “baseline CAR”? (It may help the reader who is not familiar/specialist in this field).

Response to the Reviewer: Thank you for highlighting this aspect. We used "baseline" and "pre-treatment" synonymously. We have marked this again to improve comprehensibility.

(15) Regarding Figure 1. Why CAR is evaluated as a continuous variable in this case, and not as a dichotomic using the median cut-off?

Response to the Reviewer: Understandable argument! In fact, this allows to offset imprecision of the median. It was more intuitive for us to use two continuous variables as such. Statistically, there was no difference.

(16) Since the OS and PFS analysis uses the CAR high/low, could you please correlate CAR high/low with the other clinicopathological variables as shown in table 1? Including the p values? Or do you think that the information present as a continuous CAR variable is enough?

Response to the Reviewer: As stated above, CAR was sufficient as a continuous variable. Nevertheless, no other correlations were observed in our cohort.

(17) Regarding the multivariate analysis, I understand that you performed a COX for OS, with the method of backward conditional?

Response to the Reviewer: Correct!

(18) In Table 3. “Removed” means removed in the process of backward conditional?

Independently of the other variables. Did anti-PD-L1 therapy improved the survival of the patients?

Response to the Reviewer: Correct! Consequently, all independent variables that significantly predicted the outcome in univariate Cox regression analysis (Supplementary Table S2) were first included in the model and then successively removed if they did not contribute significantly.

As mentioned above, only a minority of patients have received durvalumab as standard treatment after CRT during the observation period. Nevertheless, patients received a variety of second/third line treatments. Therefore, PFS is particularly interesting. In contrast, OS is significantly influenced by second and third line therapies.

(19) Regarding the CAR variable. Which component of it (CRP or albumin) was the most relevant for the prognosis of the patients?

Response to the Reviewer: This is an excellent question that is difficult to answer. As discussed in the manuscript, both CRP and albumin have been described as exclusive predictors of survival in patients with NSCLC in various settings. However, when considered individually, these markers are only general and affected by other factors, so they cannot be used as prognostic marker for an individual patient. Therefore, combined scores such as the CAR or the GPS are more reliable.

(20) If CRP is associated to antitumor host immune response, how do you interpret that high CAR associated to poor prognosis? Or CRP is reflecting other immune system status?

Response to the Reviewer: This is also an interesting consideration that unfortunately cannot be generalized. Actually, elevated CRP levels are associated with advanced disease, metastasis, and poor prognosis.

We would like to take this opportunity to thank you again for your extremely detailed review! As you can see, we have created a list of abbreviations and an additional supplementary table. Many of your profound and helpful comments helped to significantly improve the manuscript. However, we hope you understand that we could not consider all comments. We would like to keep the manuscript concise. Accordingly, we have been able to give extensive feedback to your comments and hope that we have been able to use them to strengthen the manuscript.

Yours sincerely

MF

Round 2

Reviewer 1 Report

The authors answered the reviewer's questions in detail, but I would recommend that they include an explanation about NLR and MLR in the discussion.

Author Response

Dear Reviewer, 

again, thank you for the opportunity to re-submit our manuscript! 

We are very pleased to present you the manuscript under consideration of your remarks (NLR/LMR) and we now hope that it entirely fulfills your expectations. 

Changes are marked in yellow. 

Kind regards 

Maximilian Fleischmann